# Dirac-source diode with sub-unity ideality factor

Gyuho Myeong [1,6], Wongil Shin[1,6], Kyunghwan Sung[1], Seungho Kim[1], Hongsik Lim[1], Boram Kim[1], Taehyeok Jin[1], Jihoon Park[1], Taehun Lee[1], Michael S. Fuhrer [2], Kenji Watanabe [3], Takashi Taniguchi [3], Fei Liu [4,5✉] & Sungjae Cho [1✉]

An increase in power consumption necessitates a low-power circuit technology to extend Moore's law. Low-power transistors, such as tunnel field-effect transistors (TFETs), negative-capacitance field-effect transistors (NC-FETs), and Dirac-source field-effect transistors (DS-FETs), have been realised to break the thermionic limit of the subthreshold swing (SS). However, a low-power rectifier, able to overcome the thermionic limit of an ideality factor ($\eta$) of 1 at room temperature, has not been proposed yet. In this study, we have realised a DS diode based on graphene/MoS$_2$/graphite van der Waals heterostructures, which exhibits a steep-slope characteristic curve, by exploiting the linear density of states (DOSs) of graphene. For the developed DS diode, we obtained $\eta < 1$ for more than four decades of drain current ($\eta_{ave\_4dec} < 1$) with a minimum value of 0.8, and a rectifying ratio exceeding $10^8$. The realisation of a DS diode represents an additional step towards the development of low-power electronic circuits.

[1] Department of Physics, Korea Advanced Institute of Science and Technology (KAIST), Daejeon, Korea. [2] ARC Centre of Excellence in Future Low-Energy Electronics Technologies, and School of Physics and Astronomy, Monash University, Clayton, Victoria 3800, Australia. [3] National Institute for Materials Science, Namiki, Tsukuba, Ibaraki 305-0044, Japan. [4] School of Integrated Circuits, Peking University, Beijing 100871, China. [5] Beijing Advanced Innovation Center for Integrated Circuits, Beijing 100871, China. [6] These authors contributed equally: Gyuho Myeong, Wongil Shin, Kyunghwan Sung. ✉email: feiliu@pku.edu.cn; sungjae.cho@kaist.ac.kr

Power consumption of integrated digital devices sets the ultimate limit to downscaling and Moore's Law[1]. Reducing power consumption has been thwarted by fundamental limits on the operating voltage set by thermionic emission[2]. For an ideal thermionic device, the dependence of current $I$ on voltage $V$ is expressed through the subthreshold swing $SS = [d\log10(I)/dV]^{-1} = (k_BT/q)\log(10) \approx 60$ mV/dec at room temperature, where $k_BT$ is the thermal energy and q is the elemental charge.

Two-dimensional (2D) van der Waals (vdW) materials[3,4] have been proposed for various schemes to overcome the thermionic limit (SS = 60 mV/dec) of metal-oxide-semiconductor field-effect transistors (MOSFETs) in nonconventional transistors such as TFETs, NC-FETs, and DS-FETs[5–14]. In particular, DS-FETs use the linear energy dispersion relationship of graphene, producing a super-exponential change in the DOS with energy[15]. As a result, DS-FETs have achieved a smaller SS than that of a MOSFET, with a large drive current[11–14].

Integration of heterogeneous electronic components on a single low-power-consumption platform is highly desirable to enable applications such as the Internet of Things (IoT). Schottky diodes are important electronic components with low operation voltage and high current[16], and have many useful applications such as rectifiers, mixers, selectors, switches, photodetectors and solar cells[16]. Although there has been considerable development of low-power transistors, steep-slope diode (or triode) rectifiers that overcome the thermionic limit ($\eta < 1$) of conventional diodes have not been proposed yet, but will be necessary for device integration with low-power transistors. Herein, we propose a DS diode as an essential element for low-power circuits. The DS injects cold electrons without a long thermal tail above the potential barrier in the channel (Supplementary Figure 1). Our DS diode consists of a graphene/MoS$_2$/graphite heterojunction, where graphene acts as a cold electron injector, whereas the graphite/MoS$_2$ interface provides a Schottky barrier for rectification. The MoS$_2$ channel was chosen because of its high-gate tunability and mobility[17]. The minimum and average values of $\eta$ for the DS diode are 0.78 and less than 1 over more than four decades of current at room temperature ($\eta_{ave\_4dec} < 1$), respectively, with a high rectifying ratio (>10$^8$).

## Results

**Characteristics of Dirac-source diode**. The DS diode device (Fig. 1a, b) consists of four components: (i) an n-type monolayer MoS$_2$ channel (Supplementary Fig. 3), (ii) a graphene DS neutral at a zero gate voltage, (iii) a graphite drain-contact to form a Schottky barrier between the graphite and monolayer MoS$_2$ for electrical rectification with a bias voltage, and (iv) metal (back, top, and control) gate electrodes to tune the Fermi levels of 2D materials. Two-dimensional van der Waals epitaxy was performed inside an Ar-filled glovebox until the heterostructure was encapsulated by hexagonal boron nitride (hBN) to avoid any contamination through air exposure or chemicals (Supplementary Fig. 2). Unlike a metal contact, a graphite contact with the monolayer MoS$_2$ forms a non-reactive clean interface[18] (Supplementary Fig. 4). Cr/Au electrodes were placed only in the region where graphite or graphene encapsulated by hBN exists.

The diode has a local top-gate, control-gate and a global back-gate. The top gate only modulates the channel of the monolayer MoS$_2$ band while the control-gate tunes the regions of the monolayer MoS$_2$ channel and part of graphene overlapped with MoS$_2$, respectively. The global back-gate affects the graphene/MoS$_2$/graphite heterostructure. The gate-dependent electrical measurements (Supplementary Fig. 5) indicate that the Dirac point of hBN-encapsulated graphene is located at $V_{BG} = +1.9$ V.

Figure 1c presents the characteristic drain current ($I_D$) versus bias voltage ($V_{bias}$) curve for the DS diode at $V_{BG} = -6$ V, $V_{CG} = 0$ V and $V_{TG} = -0.7$ V. At $V_{BG} = -6$ V, graphene is p-type. When a bias voltage is applied to the graphene, electrons are injected from the p-type graphene source into the graphite drain. Note the electrons in the graphene source contributing to the current injection should have energy above the green dotted line (Fig. 1d) which is determined by the top of the MoS$_2$ conduction band edge while not all the electrons above $E_F$ in graphene contribute to the current. The injected current density from graphene is given by:

$$J(E) \propto M_0|E - E_D|f(E - E_{FS})$$

Where $E_D$ is the Dirac point and $E_{FS}$ is the Fermi level of graphene. So, as the channel barrier gets lower than the Dirac point, availible density of states from graphene around $E = E_{top}$ ($E_{top}$ is the top of channel barrier) increases due to $M_0|E - E_D|$. So, injected current increased super-exponentially and the device works as a DS-FET. The electrical measurements reveal a nearly Ohmic graphene/MoS$_2$ contact and a Schottky barrier of the graphite/MoS$_2$ contact (Supplementary Fig. 6). When a negative back-gate voltage is applied, the Schottky barrier height increases and the device current is mainly modulated by the Schottky barrier at the interface between the graphite and monolayer MoS$_2$. Although the Ohmic contact behaviour between graphene and monolayer MoS$_2$ was observed in electrical measurements, to fully understand the band diagram at the graphene/monolayer MoS$_2$ interface and its gate dependence, further studies are needed.

The performance of a Schottky diode is mainly characterised by two figures of merit. One is the rectifying ratio, which refers to the ratio between the on and off currents ($R = \frac{I_{on}}{I_{off}}$), whereas the other is $\eta$, which is the slope representing the change in drain current with a bias voltage and can be obtained from the following Schottky diode equation:

$$I_D = I_S\left(1 - e^{qV_{bias}/\eta k_BT}\right), \tag{1}$$

where q is the elementary charge, $V_{bias}$ is the applied bias voltage, $\eta$ is the ideality factor, $k_B$ is the Boltzmann constant, T is the temperature, and $I_D$ and $I_S$ are the drain and leakage currents, respectively. Equation (1) corresponds to SS = $(\eta k_BT/e)\log(10)$ hence values $\eta < 1$ correspond to SS below the thermionic limit. The characteristic curve at a negative gate voltage in Fig. 1b exhibits rectification behaviour with $\eta < 1$ observed over more than four decades of drain current, a minimum $\eta$ of 0.78, and a large rectifying ratio (>10$^8$).

**Steep-slope switching mechanism of Dirac-source diode**. To explore the switching mechanism of the DS diode, we developed an analytical formula for the ideality factor and performed numerical device simulations (Supplementary Note 6). Both the two methods show that the ideality factor less than 1 is obtained in the DS diode due to the linear density of states of graphene. The switching slope of a diode is determined by the energy-dependent current density injected from an electrode, which is related to DOS and the distribution function. Graphene has a linear energy-dependent electronic DOS near the Dirac point, which is different from conventional metals with a constant DOS around the Fermi level. Therefore, the thermal tail of the Boltzmann distribution function is suppressed by the Dirac point tuned to the off-state region by doping. Namely, as the bias voltage is decreased on the graphene electrode as shown in Fig. 1d, the part of current density related to the distribution function is increased exponentially similar to conventional

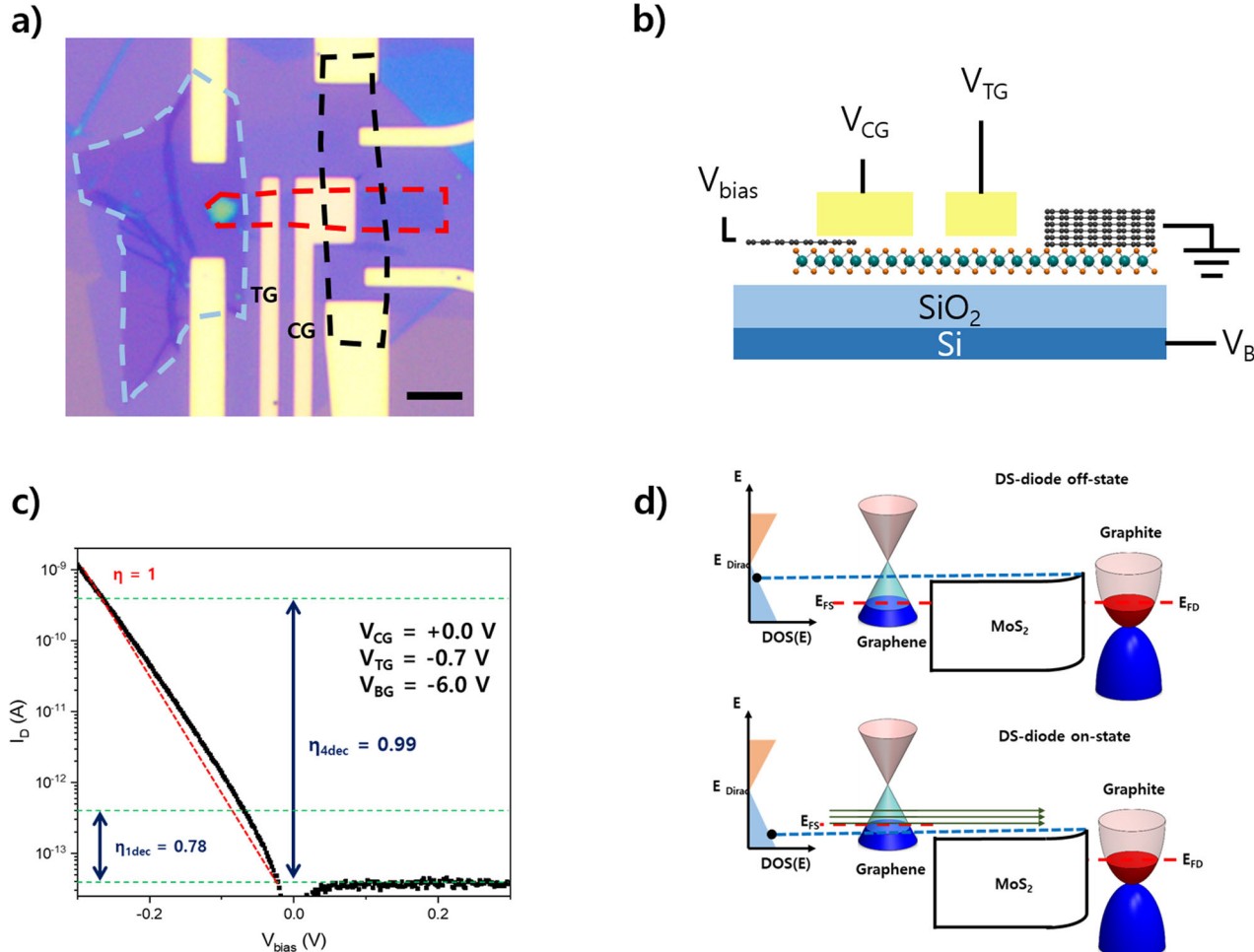

**Fig. 1 Device structure, characteristic curve, and band diagram of DS diode. a** Optical image of graphene/MoS$_2$/graphite heterojunction diode. Grey, red, and black dashed lines indicate graphite, monolayer MoS$_2$, and graphene, respectively. We used graphene as a source and graphite as a drain. The top-gate(TG) and control-gate(CG) were placed for gate modulation of the MoS$_2$ channel and graphene/MoS$_2$ overlapped region, respectively. Scale bar, 5 um. **b** Schematic image of graphene/MoS$_2$/graphite heterojunction diode. **c** Characteristic drain current($I_D$)-bias voltage($V_{bias}$) curve in our device, which exhibits ideality factor(η) = 0.78 in 1 decade of current and an average η < 1 in more than four decades of current, i.e., η$_{ave\_4dec}$ < 1. The rectifying ratio of our device is larger than 10$^8$. **d** Band diagram of DS Schottky diode, which explains the working principle of cold electron injection from graphene. E$_{Dirac}$, DOS, E$_{FS}$, and E$_{FD}$ indicate Energy at the Dirac point, the density of states, Fermi level at the source side, and Fermi level at the drain side, respectively. Blue dashed line and green arrows indicate MoS$_2$ energy window level and expression of rapid increment of current flow.

metals, which results in the ideal factor limit of 1. While, the injected DOS over the top of the channel barrier is also increased linearly from off-state to on-state, as shown in Fig. 1d. Therefore, the current is increased super-exponentially and the ideal factor below 1 is obtained in the diode with graphene electrode as the injection source.

Therefore, the switching slope of a diode, i.e. η < 1, is obtained in the diode with a graphene electrode as the cold electron injection source because of the linear DOS of the DS. Detailed simulation results are presented in Supplementary Fig. 7. Quantum transport simulations show that the DS diode has promising device performance. The ideality factor as small as 0.69 is obtained in the simulated DS diode and is less than 1 in more than five decades of current at room temperature.

The on-state current is larger than 10$^3$ μA/μm and the rectifying ratio is over 10$^7$.

**Properties of asymmetric graphene/MoS$_2$ and graphite/MoS$_2$ contacts**. Figure 2a presents the $I_D$-$V_{bias}$ characteristic curve of the DS diode at different back-gate voltages. For the DS diode to work

as a diode, an asymmetric Schottky barrier height between the source and drain is necessary[19–22]. To satisfy this condition, we placed asymmetric graphene and graphite contacts with the monolayer MoS$_2$ channel with gates. Without gate modulation, graphene has a work function of 4.3–4.7 eV from a monolayer to a few layers[23–25]. Because the work function of graphene (~4.3 eV) does not differ significantly from the electron affinity of MoS$_2$ (~4.2 eV)[26–29], the Schottky barrier height at the graphene/MoS$_2$ interface is negligible, compared to the Schottky barrier height at the graphite/MoS$_2$ interface. This also indicates that the Dirac point of pristine graphene is located near the conduction band edge of MoS$_2$. As shown in Supplementary Fig. 10, in case of the metal/n-type semiconductor junction, the positive voltage on metal became forward bias. In our case, we applied bias voltage on the graphene side, and negative bias became forward bias, i.e., positive bias on the graphene side is forward bias, which indicates the Schottky barrier between the graphite/MoS$_2$ junction is dominated in our device. Supplementary Fig. 6 indicates that the graphene/MoS$_2$ device shows an almost Ohmic IV curve, whereas graphite/MoS$_2$ does not show an Ohmic IV curve at room temperature. Figure 2a shows that as the gate voltage decreases, the rectification behaviour

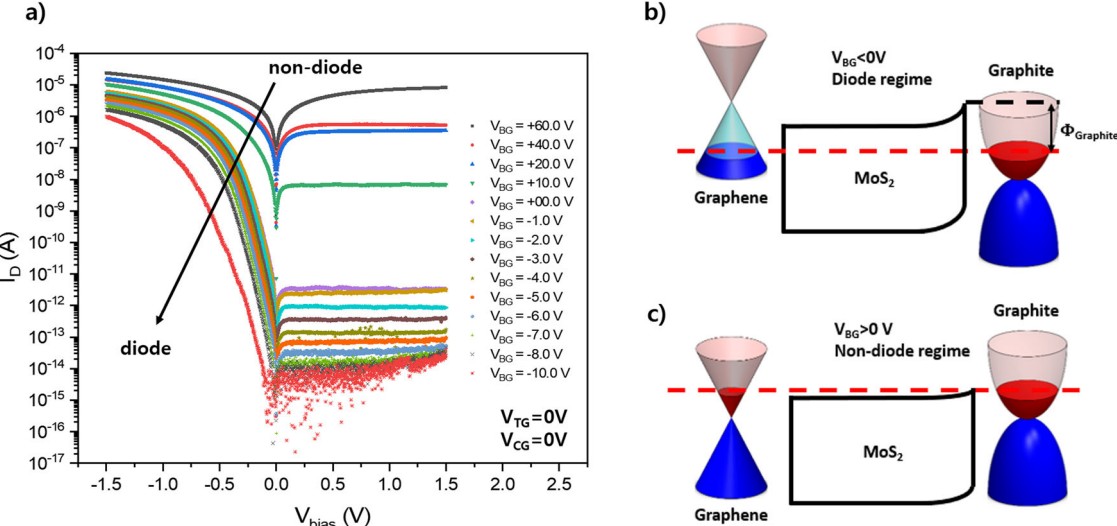

**Fig. 2 Characteristic $I_D$-$V_{bias}$ curve for various $V_{BG}$ and its band diagram. a** Characteristic $I_D$-$V_{bias}$ curve in the range of $V_{BG} = -10$ to $+60$ V. As $V_{BG}$ decreases, change from non-diode to diode behaviour is observed. **b** Band diagram when $V_{BG} < 0$ (diode regime). Owing to the larger work function of graphite than that of graphene, the device becomes a graphite/$MoS_2$-interface Schottky barrier-dominant Schottky diode. **c** Band diagram when $V_{BG} > 0$ (non-diode regime). As $V_{BG}$ increases, the work function of graphite decreases, and the Schottky barrier height of the graphite/$MoS_2$ interface decreases.

becomes dominant at negative gate voltages. As the back-gate voltage exceeds $V_{BG} > 0$, non-diode $I_D$-$V_{bias}$ characteristic curves appear.

To clarify the origin of the gate-dependent modulation of the $I_D$-$V_{bias}$ characteristic curves, we measured the modulation of the Schottky barrier height with back-gate voltages from the activation energy in the reverse bias regime. The Schottky diode equation (Eq. 1) can be rewritten as

$$I_D = AA^*T^{\alpha}e^{-q\Phi_B/k_B T}\left(1 - e^{\frac{qV_{bias}}{\eta k_B T}}\right), \quad (2)$$

where $A$ is the area of the Schottky junction, $A^*$ is the Richardson constant, $\alpha = 3/2$ is an exponent for a two-dimensional semiconducting system[30], $k_B$ is the Boltzmann constant, q is the elementary charge, T is the temperature and $\Phi_B$ is the Schottky barrier height. When a large negative bias in absolute value is applied, i.e. $e^{qV_{bias}/k_B T} \approx 0$, the saturated drain current is proportional to $T^{3/2}e^{-q\Phi_B/k_B T}$. The inset of Supplementary Fig. 11a shows a plot of $\ln(I_{sat}/T^{3/2})$ versus $1/k_B T$ in the reverse bias saturation regime ($V_{bias} = +1$ V). We extract $\Phi_B$ for a given $V_{BG}$ from the slope of each curve. Supplementary Fig. 11a shows the Schottky barrier height obtained from the slope of each curve in the inset of Supplementary Fig. 11a. As shown in Supplementary Fig. 11b, in the highly positive $V_{BG}$ regime, the device shows an almost linear $I_D$-$V_{bias}$ curve, exhibiting nearly Ohmic contact behaviour (negligible Schottky barriers on both sides of the contacts, graphene and graphite with $MoS_2$).

**Dirac-source field-effect transistor measurement.** To prove that the proposed diode is operated via cold carrier injection from a graphene DS at negative back-gate voltages, we measured the SS to determine if it showed sub-thermionic values. Supplementary Fig. 12b shows the characteristic $I_D$ versus control-gate voltage ($V_{CG}$) transfer curve under the working conditions of the DS-FET, i.e. $V_{BG} < 0$ V, where the graphene is p-type. When we apply $V_{BG} = -3$ V, graphene slightly p-type. When the control-gate is placed on the $MoS_2$ channel and the graphene/$MoS_2$ overlapped region is swept from the off-state to the on-state, the DOS of the graphene increases according to the band diagram presented in Supplementary Fig. 12a, thereby operating as a DS-FET. As shown in Supplementary Fig. 12b, the $SS_{ave\_1dec}$ and $SS_{ave\_3dec}$ exhibits

53.6 and 58.75 mV/dec, respectively, which indicates that the proposed diode acts as a DS-FET owing to the linear energy dispersion relationship of the graphene-source electrode, resulting in a super-exponential change in the DOS. Both DS-FET and DS diode have the same origin for SS < 60 mV/dec and $\eta$ < 1.

**Steep-slope diode curves in the p-doped graphene region.** Figure 3 shows the $I_D$-$V_{bias}$ characteristic curve in the steep-slope diode regime at $V_{BG} = -6$ to $-2$ V in 2 V step with fixed top- and control- gate voltages ($V_{TG} = -0.7$ V and $V_{CG} = 0$ V), where the graphene is p-doped. In the measured regime, where the top of the Schottky barrier is located below the Dirac point of graphene, $\eta$ of the device is less than 1 in more than four decades of current owing to the cold charge injection from the DS at a forward bias ($V_{bias} < 0$). The minimum $\eta$ that we measured in one decade of current is 0.78. The red dotted line in Fig. 3 is an ideal diode curve ($\eta = 1$) in the forward bias direction. The DS diodes in these gate voltage regions show rectification ratios exceeding $10^8$ at $V_{BG} = -6$ V (>$10^6$ when $V_{BG} = -2$ V and >$10^7$ when $V_{BG} = -4$ V). We note that the device leakage current level is limited by the leakage currents (~10 fA) from the measurement equipment. Therefore, the reverse bias leakage current level from the diode should be lower than the measured values.

**Discussion**

In conclusion, we successfully demonstrated the DS diode that operates based on cold charge injection from a graphene source owing to the linear DOS and a Schottky barrier at the interface between graphite and monolayer $MoS_2$. As the linear DOS of the injected charges from p-type graphene over the top of the Schottky barrier between graphite and n-type monolayer $MoS_2$ increases linearly from reverse to forward bias, an ideal factor below 1 is obtained in the diode with a graphene electrode as the injection source. Using gate modulation of the Schottky barrier height of the graphite/$MoS_2$ junction, gradual switching between the diode and non-diode behaviours was also observed. The fabricated DS diode presents a minimum $\eta$ as low as 0.78 in one decade of current, and it remains less than 1 for more than four decades of current at room temperature ($\eta_{ave\_4dec} < 1$), with a high rectifying ratio exceeding $10^8$. Additionally, the device shows SS < 60 mV/dec for the same origin as that for $\eta$ < 1. By using CVD-grown $MoS_2$, graphene and graphite,

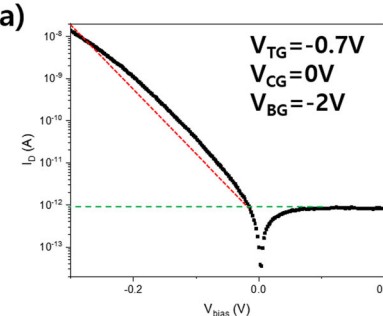 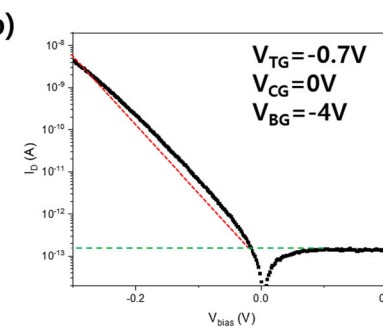 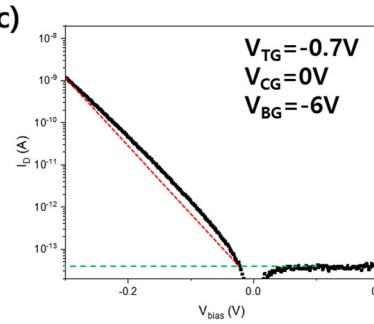

**Fig. 3 Slopes of DS Schottky diode versus ideal diode and recorded ideality factor in 2D vdW material-based diode.** Comparison of slopes between the DS Schottky diode and an ideal diode. Black and red dotted data represent those of the DS Schottky diode and an ideal diode, respectively. The Green dashed line indicates off-state current in the reverse bias regime. **a** DS Schottky diode curve at $V_{BG} = -2$ V. **b** DS Schottky diode curve at $V_{BG} = -4$ V. **c** DS Schottky diode curve at $V_{BG} = -6$ V. The DS Schottky diode exhibits a $\eta_{ave\_3dec}$ of 0.98, 0.95, 0.94 when $V_{BG} = -2, -4$, and $-6$ V, respectively with fixed top- and control- gate voltage of $V_{TG} = -0.7$ V and $V_{CG} = 0$ V.

integrated circuits using steep-slope DS-FETs and DS diodes can be fabricated in a large scale and pave the way for energy-efficient circuit technology.

## Methods

**Device fabrication.** Supplementary Fig. 2 illustrates the fabrication of the Dirac-source (DS) Schottky diode. As can be seen, the first step involves the preparation of a polydimethylsiloxane (PDMS) stamp covered with a polycarbonate (PC) film on a glass slide. Subsequently, $MoS_2$ flakes are mechanically exfoliated on a Si/SiO$_2$ wafer. In this study, the $MoS_2$ exfoliation was performed in an Ar-filled glovebox to prevent contamination. Using the standard dry-transfer method, each flake is picked up in the order—top hexagonal boron nitride (hBN), graphite, graphene, $MoS_2$, and bottom hBN. After fabrication of the PC film and confirming sufficient adherence of the prepared flakes, the wafer was slowly heated to 90 °C, during which time, the sliding glass is slowly raised. During the pick-up process, owing to the large area of the top hBN, graphene, graphite, and $MoS_2$ do not directly touch the PC film. After fabrication of the heterostructure on the PC film, the latter is slowly placed onto a prepared 285-nm-thick Si/SiO$_2$ wafer. Subsequently, the wafer is heated to 180 °C, thereby melting the PC film. Thereafter, the PC film is successively washed using chloroform, acetone, and isopropyl alcohol (IPA). After transfer of the heterojunction to a new wafer, the device is exposed to chemicals to erase the released PC film. However, graphene, graphite, and $MoS_2$ layers are encapsulated within large areas of the top and bottom hBN layers, which the chemicals cannot percolate. After fabricating the heterostructure on a 285-nm Si/SiO$_2$ wafer, the standard e-beam lithography and plasma etching procedures are performed via e-beam deposition (Cr/Au = 5/60 nm) to place electrical contacts in the graphene and graphite layers. One-dimensional edge contact on graphene was formed in this process[31]. The hBN and graphite layers are etched using CF$_4$/O$_2$ and Ar/O$_2$, respectively. Additional e-beam lithography and deposition processes are performed to facilitate top- and control-gate placement.

**Measurement.** Supplementary Fig. 13 depicts the measurement protocol of the DS diode. Using the Keithley 6430, bias voltage was applied to the graphene electrode and measured the drain current from the graphite electrode. Keithley 2400 was used to apply a gate voltage to the Si back-gate electrode ($V_{BG}$) and two Yokogawa 7651 were used to apply gate voltages to the top- and control-gate electrodes ($V_{TG}$ and $V_{CG}$, respectively). Measurements were performed in a vacuum probe station with tri-axial cables to reduce leakage current from the measurement setup.

## Data availability

Relevant data supporting the key findings of this study are available within the article and the Supplementary Information file. All raw data generated during the current study are available from the corresponding authors upon request. Source data are provided with this paper.

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

## Acknowledgements

We thank J. Lee for the helpful discussions. S.C. acknowledges support from Korea NRF (Grant Nos. 2020M3F3A2A01081899, and 2020R1A2C2100258). F.L. acknowledges support from NSFC (Grant No. 61974003) and the 111 Project (Grant No. B18001). M.S.F. acknowledge support from the ARC (CE17010039).

## Author contributions

S.C. conceived and supervised the project. G.M., W.S. and K.S. fabricated devices and performed measurements. K.W., and T.T. grew high-quality hBN single crystals. S.K., J.P., K.S., H.L., B.K., T.J., and T.L. assisted high-temperature transport measurements. F.L. developed the theoretical model and performed device simulations. S.C., G.M., W.S., M.S.F., and F.L. analyzed the data. S.C., and G.M. wrote the manuscript. All the authors contribute to editing the manuscript.

## Competing interests

The authors declare no competing interests.
