## [Peer Review File · Nature Communications]

Dirac-Source Diode with Sub-unity Ideality FactorREVIEWER COMMENTS

Reviewer #1 (Remarks to the Author):

In this work, Myeong et al. reported an asymmetric MoS₂ diode structure contacted by graphene and graphite. They exploit the tunable DOS of graphene which acts a cold source of electron injection to achieve sub-unity ideality factor in the diode. The concept of the device is similar to those reported in CNT and 2D semiconductor FETs (Ref. 7-10) and the main novelty is the use of asymmetric contacts to realize diode behavior. The demonstrated DS diode is an expansion of library of low-power switches in future electronics. However, I have several major concerns regarding the device mechanism and generality of this approach. Before further consideration the following points should be addressed and clarified, and the more experiments are suggested to substantiate the conclusions.

1. I find the use of G1 and G2 regions very confusing in the device structure. The authors include these regions in several figures but it's not clear how they contribute to the sub-unity ideality factor. Does the G1/G2 homojunction contribute to the device characteristics described in this paper? The authors should make this point clear. In their simulation in Fig. S7, the graphene region does not have different doping levels.
2. The mechanism of device operation is unclear and requires in-depth discussions. The authors only observe sub-unity ideality factor in the hole doped region, how about electron doped region ($V_g > 0$)? If the steep switching is indeed due to the tuning of Dirac point by bias voltage, electron doped region should also show similar behavior. Furthermore, more simulation results are needed to study different regimes in a comprehensive way.
3. The FET mechanism in Fig. S8 requires more clarification. The authors state that "When the top gate placed on the MoS₂ channel is swept from the off state to the on state, the DOS of the graphene increases according to the band diagram presented in Fig. S8c, thereby operating as a DS-FET." This statement is not correct because the topgate does not overlap with graphene, so the doping of graphene should not change as the topgate sweeps. I don't see how this device structure could achieve sub-60 SS. If we look at Fig. S8 carefully, the sub-60 points are all located at very low current level, which can introduce a lot of uncertainties in data analysis.
4. The authors need to demonstrate the generality of this approach to other 2D materials to fit the DS diode in a broader and more relevant context. What happens if they change the channel material to p-type semiconductors? Even further, what happens if they change graphite to metal contact? The use of graphite as contact is not scalable for device fabrication.
5. The thermionic behavior in diode and FET is temperature dependent. In order to demonstrate the breaking of thermionic limit in a more convincing manner, low-temperature measurements are suggested.
6. The authors state that "The adjustable Schottky barrier height with gate voltage indicates that Fermi-level pinning does not exist at the interface between graphite and monolayer MoS₂." This is not correct because even with Fermi-level pinning, it is still possible to tune the SBH, but with a non-unity slope (see Ref. 18).

Reviewer #2 (Remarks to the Author):

The idea of the manuscript is good, but unfortunately the writing and the figures are quite unclear.

1. The fabrication process is not clear. For example, Figure S2: how do the authors make sure that the Cr/Au electrodes are touching the graphene and graphite? It is not clear from Figure (h)-(i) at all.
2. Figure 1b: What is VTG held at during this measurement? At VBG = -30 V, is the MoS₂ still ON? At VBG = -30 V, holes are dominant in graphene. So holes should move from graphene to graphite through MoS₂. The authors mention that electrons are being injected from graphene to graphite. Please elaborate.
3. Figure 2: Again, what is VTG here?

4. Figure 3: I am not sure how the authors are claiming that an ideality factor of less than 1 is sustained for >2 decades. By the green line drawn, it is clear that ideality factor of less than 1 is maintained for a little over 1 decade for VBG = -15 V, -30 V and -45 V.

5. How repeatable are the I-V measurements? Will the ideality factor be retained for 10 consecutive sweeps? Please include comments in the manuscript.

6. Will the device behave the same way if one were to use CVD-grown MoS2 and graphene? Please include comments in the manuscript.

Reviewer #3 (Remarks to the Author):

The manuscript entitled "Dirac-Source Diode with Sub-unity Ideality Factor" have demonstrated a Dirac Source diode, which exhibits a steep-slope characteristic curve by using graphene electrodes. This topic and their findings are very interesting. However, one of the major problems is the large leakage current for I_{ds} - V_{ds} measurement. As shown in Fig. 1b and Fig. 3, the I_{ds} current cross zero (y -axis) at very negative V_{ds} voltage, indicating the leakage current have been coupled into the I_{ds} measurement. Therefore, the measured I_{ds} may not truly reflect the diode behavior. This could be important because the I_{ds} current (of sub-1-ideality factor region) is only one order higher than the leakage current. To ensure the accurate ideality factor, lower leakage current (below pA) or higher resolution equipment need to be used, which should be satisfied by standard semiconductor analyzer.

Similarly, if we look into the data in Fig. S8a, b, the top gate leakage current also have a strong influence to the off-state current, which could be another reason for the measured $SS < 60$ mV/dec (which also happens at the device off state). Hence, this top gate leakage should also be minimized to support the author's claim.

Based on the present data, I can not support for its publication for Nature Communications. There are some other minor questions as below

1. The device optical image should be included in Fig. 1.

2. The diode has a local top gate, but the fabrication processes are not described in the manuscript.

3. At negative back gate voltage, large Schottky barrier should exist between p-type graphene and n-type MoS2, as claimed by the author "When a negative back-gate voltage is applied, the Schottky barrier height increases". However, the author also claim the device is dominated by the barrier at graphite and MoS2: "the device current is mainly modulated by the Schottky barrier at the interface between the graphite and monolayer MoS2". This point should be more clarified.

4. Similar as the previous question, the diode behavior only exists at negative back gate voltage according to Fig. 2. Because the graphene/MoS2 barrier is more sensitive to the gate voltage while graphite/MoS2 barrier is nearly insensitive to the gate voltage, does this behavior suggest the observed behavior is more governed by the graphene/MoS2 rather than the proposed graphite/MoS2 junction?

5. For band diagram in Fig. 2b, the graphene should have large Schottky barrier with MoS2. For diagram in Fig. 2c, if the MoS2 is degenerated, the band bending is confusing at graphite part. What is the work function of graphite used here?

Reviewer #1 (Remarks to the Author):

In this work, Myeong et al. reported an asymmetric MoS₂ diode structure contacted by graphene and graphite. They exploit the tunable DOS of graphene which acts a cold source of electron injection to achieve sub-unity ideality factor in the diode. The concept of the device is similar to those reported in CNT and 2D semiconductor FETs (Ref. 7-10) and the main novelty is the use of asymmetric contacts to realize diode behavior. The demonstrated DS diode is an expansion of library of low-power switches in future electronics. However, I have several major concerns regarding the device mechanism and generality of this approach. Before further consideration the following points should be addressed and clarified, and the more experiments are suggested to substantiate the conclusions.

1. I find the use of G1 and G2 regions very confusing in the device structure. The authors include these regions in several figures but it's not clear how they contribute to the sub-unity ideality factor. Does the G1/G2 homojunction contribute to the device characteristics described in this paper? The authors should make this point clear. In their simulation in Fig. S7, the graphene region does not have different doping levels.

Our reply: We thank the Referee for bringing up this point. In fact, the important part of graphene to act as a cold source of electron injection is both G₁ and G₂. Since the channel of the FET (MoS₂) is n-type, graphene source of both G₁ and G₂ should be p-type as shown in the main Figure 1 (below) of previous manuscript for the graphene to inject cold electrons into the channel (see Supplementary section 1).

The difference in G_1 and G_2 arises since the doping of bare graphene is different from the doping of graphene on top of MoS₂ as shown in the supplementary Figure S5 of previous supplementary manuscript (above). As V_{BG} changes, the resistance of graphene, i.e. both parts of bare graphene and graphene on top of MoS₂ shows two peaks as in Figure c) above. However, in our new device, the doping of graphene due to MoS₂ was negligible (see below). It seems that the doping level changes from sample to sample depending on the initial doping of bulk MoS₂ crystal we use to exfoliate monolayer MoS₂. Therefore, we have now removed the G_1 and G_2 and instead put integrated region G (graphene), which again should be p-type for cold charge injection into n-type MoS₂ channel. We have modified the main Figure 1 and Supplementary Figure S5 as below.

Supplementary Figure. S5 (gate-dependent resistance of graphene across MoS₂)

Main Figure. 1

In the revised manuscript, we studied different doping levels on switching properties of DS diode by simulations as shown in Fig. S7 and S8 as below and added a discussion at line 1, page 7 of the Supplementary Information: “We also studied the impact of the doping level of graphene on switching properties of DS diode as shown in Fig. S7(b). The ideality factor of DS diode with p-type graphene is less than one at the bias voltage region between -0.1 V and -0.3 V, and the current is increased over four orders of magnitude. While, ideality factor gets larger than one as graphene is intrinsic or n-type as shown in Fig. S8(a), because the Dirac point is always below the top of the channel barrier and cannot filter high energy thermionic current. Fig. S8 (b) shows that there is an obvious phase transition of ideality factor from sub-unity to over-unity as graphene is doped from p-type to n-type.”

Figure. S7

Figure. S8

2. The mechanism of device operation is unclear and requires in-depth discussions. The authors only observe sub-unity ideality factor in the hole doped region, how about electron doped region ($V_g > 0$)? If the steep switching is indeed due to the tuning of Dirac point by bias voltage, electron doped region should also show similar behavior. Furthermore, more simulation results are needed to study different regimes in a comprehensive way.

Our reply: As shown in the main Figure 1 of the new (and old) manuscript, the graphene region should be p-type for the graphene source to inject cold electrons into the n-type diode channel (MoS_2). The

mechanism of how sub-unity diode behavior occurs is as follows: As the bias voltage applied to graphene increases to positive direction (reverse bias) from negative forward bias (main Figure 1c)), the electrochemical potential of graphene electrode decreases due to the negative electron charge. Therefore, the energy above which electrons inject becomes closer to the Dirac point as the bias voltage increases (to reverse bias) and the diode becomes off-state. Due to the linear DOS, this super-exponential decrease of carrier density (please see Supplementary Figure 1) allows steeper slope of the diode on-off transition in our DS Schottky diode than the conventional ideal Schottky diode.

In the revised manuscript, we performed more calculations to study different operation regimes and added a discussion at line 8, page 7 in the Supplementary Information: “Next, we studied electron doped region of graphene to realize sub-unity ideality factor. There are two important factors in achieving an ideality factor less than one in a DS diode: the doping type of graphene and the Schottky barrier height between graphite and ML MoS₂. In order to realize sub-unity ideality factor in electron doped region of graphene, a negative gate voltage has to be applied to achieve carrier transport by valence band of ML MoS₂. We first fix the Schottky barrier height ($\Phi_B = 0.48$ eV) between graphite and ML MoS₂ as that in hole doped region of graphene, and apply n-type graphene with p-type Ohmic contact between graphene and ML MoS₂. Fig. S9 (a, b) show such device cannot reach ideality factor less than one when current is larger than 1×10^{-10} $\mu\text{A}/\mu\text{m}$, because there is a larger p-type Schottky barrier height between graphite and ML MoS₂. The current is mainly tunneling current through the p-type Schottky barrier and the Dirac point of graphene is not decisive to the transport as shown in Fig. S9 (c). If the Schottky barrier height is set at $\Phi_B = 1.17$ eV, sub-unity ideality factor can be obtained in DS diode with electron doped graphene as shown in Fig. S9 (a, b). At $V_{\text{bias}} = 0.15$ V, the Dirac point of graphene gets lower than the top of channel barrier and can filter thermionic carriers over the barrier for sub-unity switching as shown in Fig. S9 (d).”

Fig. S9

3. The FET mechanism in Fig. S8 requires more clarification. The authors state that “When the top gate placed on the MoS₂ channel is swept from the off state to the on state, the DOS of the graphene increases according to the band diagram presented in Fig. S8c, thereby operating as a DS-FET.” This statement is not correct because the top gate does not overlap with graphene, so the doping of graphene should not change as the top gate sweeps. I don’t see how this device structure could achieve sub-60 SS. If we look at Fig. S8 carefully, the sub-60 points are all located at very low current level, which can introduce a lot of uncertainties in data analysis.

Our reply: Below is the new data set we obtained from our additional experiment of measuring a new MoS₂ Dirac Source triode in a new measurement setup with triaxial cable (new version of Figure S10). Here we show the I_D vs control gate (instead of top gate) which has overlap with graphene. As shown in the band diagram below, as the DS FET turns from on-state to off-state, the DOS of graphene outside the control gate region decreases which agrees well with previous reports of DS FET operation (Qiu, C. *et al.* Dirac-source field-effect transistors as energy-efficient, high-performance electronic switches. *Science* 361, 387-392 (2018)). Please note that the electrons in graphene source contributing to the current injection should have energy above green dotted line in the band diagram which is determined by MoS₂ conduction band edge while **not all the electrons in the energy range $E_F < E < E_F + eV_{\text{bias}}$ in graphene** contribute to the current. The injected current density from graphene is given by:

$$J(E) \propto M_0 |E - E_D| f(E - E_{FS})$$

Where E_D is the Dirac point and E_{FS} is the Fermi level of graphene. So, as the channel barrier gets lower than the Dirac point, available density of states from graphene around $E = E_{\text{top}}$ (E_{top} is the top of channel barrier conduction band edge of MoS_2) increases due to $M_0 |E - E_D|$. So, injected current increased super-exponentially and the device works as a DS-FET.

Therefore, this sub-60 SS could also be achieved by top gate which does not cover graphene. Again, **the cut off energy above which electrons inject into the MoS_2 channel is not determined by graphene E_F , but determined by the top of conduction band edge of MoS_2 .** The top gate tunes the Fermi level (or band edge) only in the MoS_2 channel. Although the Fermi level of graphene is set as a constant value (p-type doping by back-gate voltage = -3V), carriers in graphene only above the conduction band edge of MoS_2 contribute to conduction. As the conduction band edge increases from on state to off state, the Fermi level of graphene (crossing the conduction band edge of MoS_2 and above which contribute to the conduction) becomes closer to the Dirac point. In this case, due to the linear decrease of graphene DOS from on to off state (thereby super-exponential decrease in carrier density of graphene source), the SS becomes smaller than 60mV/dec .

To make the above statement clarified in the manuscript, we add a phrase in line 64 on page 3 (main manuscript) “Note the electrons in graphene source contributing to the current injection should have energy above green dotted line (Fig. 1d) which is determined by the top of MoS_2 conduction band edge while not all the electrons above E_F in graphene contribute to the current. The injected current density from graphene is given by:

$$J(E) \propto M_0 |E - E_D| f(E - E_{FS})$$

Where E_D is the Dirac point and E_{FS} is the Fermi level of graphene. So, as the channel barrier gets lower than the Dirac point, available density of states from graphene around $E = E_{CBM}$ (E_{CBM} is the conduction band edge of MoS_2) increases due to $M_0 |E - E_D|$. So, injected current increased super-exponentially and the device works as a DS-FET.”

4. The authors need to demonstrate the generality of this approach to other 2D materials to fit the DS diode in a broader and more relevant context. What happens if they change the channel material to p-type semiconductors? Even further, what happens if they change graphite to metal contact? The use of graphite as contact is not scalable for device fabrication.

Our reply: As the referee points out, we tried to demonstrate the DS diode with p-type transition metal dichalcogenide such as WSe_2 or MoTe_2 . However, we have realized that those p-type materials cannot be used to realize p-type DS Schottky diode **at least in the device structure we use**. We might be able to devise a new device structure in the future to realize p-type DS diode, but this work is beyond the current research we report here.

The conditions which graphene and p-type TMDC should satisfy simultaneously are as follows: (1) graphene should be n-type (since the channel is p-type) => this means that positive V_{BG} should be applied because charge neutrality point of graphene is usually near $V_{BG}=0\text{V}$, (2) at positive V_{BG} the channel should be p-type. We can make the graphene p-type by applying $V_{BG}>0$, however as far as we know, no 2D materials are p-type at positive gate voltage (please see the figures below from different references). Therefore the conditions for p-type DS Schottky diode cannot be satisfied with any p-type 2D materials we can use.

NPG Asia Materials volume 10, pages703–712 (2018)- WSe2

Adv. Mater., 26: 3263-3269. (MoTe2)

We have fabricated Schottky diode with graphene and p-type TMDC WSe₂. The characteristic curve is as follows.

$V_{bg} = -45.0 \text{ V} @ V_{cg} = +0.5 \text{ V} \ \& \ V_{tg} = +0.0 \text{ V}$

Here the graphene and graphite were used to contact p-type WSe_2 . We had to apply negative gate voltage, otherwise the channel becomes either n-type or inside a bandgap (see the bottom graph above). We could only observe nearly ideal Schottky diode behavior with ideality factor slightly greater than 1 at negative gate voltages, where graphene is p-type while channel is p-type. Again, for this p-type Schottky diode to perform as DS diode, the graphene should be n-type and the channel should be p-type. Both conditions cannot be met simultaneously since negative gate voltage should be applied for WSe_2 to be p-type while positive gate voltage should be applied for graphene to become p-type.

One could doubt why the graphene and MoS_2 cannot be gated separately. However, it is physically almost impossible to put two gates to make all the graphene region n-type and all the MoS_2 region p-type at the interface between them. In a real device, even if two gates are separately placed on top of MoS_2 and graphene, they affect the other regions as well.

In principle, metal contact having high work function should form Schottky barrier to n-type MoS_2 similar to graphite contact. However, deposition of metal contacts onto the 2D materials has been reported to have invasive effects (Liu, Y. *et al.* Approaching the Schottky-Mott limit in van der Waals metal-semiconductor junctions. *Nature*. 557, 696-700 (2018)). Thus we tried to avoid this problem by using graphite rather than metal contacts. We agree that achieving scalable (while not invasive) contacts for MoS_2 is a very important problem although it is beyond the scope of our research.

5. The thermionic behavior in diode and FET is temperature dependent. In order to demonstrate the breaking of thermionic limit in a more convincing manner, low-temperature measurements are suggested.

Our reply: We used the term “thermionic behavior in conventional diode” by meaning as follows:

(please also see Supplementary Figure 1 below)

The continuous density of states (DOS) of a normal metallic source exhibits the Boltzmann distributed electron density given by $n(E) \approx \exp((E_F - E)/k_B T)$. For a Dirac source, the linearly varied DOS produces a super-exponentially decreasing electron density (given by $n(E) \approx (E_{Dirac} - E) \exp((E_F - E)/k_B T)$) with decreasing energy. Therefore, carrier distribution in the source of DS diode and FET has steeper super exponential decays with energy, **overcoming thermionic exponential distribution of carrier density** in conventional diode and FET. Therefore, **the ideality factor and subthreshold swing of our DS diode and DS FET can have lower value than the conventional limits (ideality factor = 1 and subthreshold swing = 60mV/dec at 300K), which again originates from thermionic carrier distribution of the source in the conventional diode and FET.** To clarify the term “thermionic limit”, we add the above statement in the Supplementary Section 1. “Therefore, carrier distribution in the source of DS diode and FET has steeper super-exponential decays with energy, **overcoming thermionic exponential distribution of carrier density** in conventional diode and FET. Therefore, **the ideality factor and subthreshold swing of our DS diode and DS FET can have lower value than the conventional limits (ideality factor = 1 and subthreshold swing = 60mV/dec at 300K), which again originates from thermionic carrier distribution of the source in the conventional diode and FET”**

As the referee suggested, we have additionally performed temperature-dependent measurements. To decrease noise level (off state current level in diode measurement) from ~ 50 pA to < 10 fA (three orders of magnitude lower noise level), we newly set up the measurement equipment by replacing all the measurement cables with triaxial cables. In the new system, we cannot cool the device, but can heat up the device up to 350K. We have now included the temperature-dependent measurement in Supplementary Section 13 with Supplementary Figure. S13. We observed that ideality factor decreases with temperature, which can be understood by the analytical formula Eq. (4) in the Supplementary Information. In a conventional ideal Schottky diode, an unity ideality factor does not change with temperature, since the current $I = I_0(1 - \exp(-\frac{V_{bias}}{k_B T}))$. The ideality factor of DS diode is given

by: $\eta = 1 - k_B T / (E_D - E_{top})$. The energy difference between E_D and E_{top} does not change with temperature and is much larger than $k_B T$. So, the ideality factor gets smaller as the increasing of temperature as shown in Figure. S16(c). The measured temperature dependence of ideality factor can be well described by the formula (the red solid line) in Fig. S16(b). It should be noted that ideality factors can only be compared at the same temperature: the smaller the ideality factor, the better switching performance. Sub-threshold swing of DS diode increases with temperature, which is consistent with that of DS FET. In the revised supplementary information section 14, we added a discussion about temperature dependence of ideality factor as follows:

“Figure. S16(a) shows temperature dependent DS-diode measurement from 300K to 350K at fixed gate voltages $V_{CG}=0V$, $V_{TG}=-0.7V$, and $V_{BG}=-6V$. We observe that averaged ideality factor decreases with temperature in Figure. S16(b). The ideality factor of DS diode is simplified as follows according to Eq. (4) :

$$\eta = 1 - \frac{k_B T}{E_D - E_{top}} \quad (7)$$

Where $E_D - E_{top} > nk_B T$. The energy difference E_D and E_{top} between does not change and the ideality factor gets smaller as the increasing of temperature as shown in Figure. S16(b). The measured temperature dependence of ideality factor can be well described by the formula (the red solid line) in Figure. S16(b). It should be noted that ideality factors can only be compared at the same temperature: the smaller the ideality factor, the better switching performance. The decreasing of ideality factor as a function of temperature does not mean the improvement of switching properties.

Fig. S15

6. The authors state that “The adjustable Schottky barrier height with gate voltage indicates that Fermi-level pinning does not exist at the interface between graphite and monolayer MoS₂.” This is not correct because even with Fermi-level pinning, it is still possible to tune the SBH, but with a non-unity slope (see Ref. 18)

Our reply: We appreciate the Referee correcting an error in the manuscript. We erased the statement in our manuscript since we agree that our measurement of adjustable SBH with gate voltage does not indicate absence of Fermi level pinning.

Reviewer #2 (Remarks to the Author):

The idea of the manuscript is good, but unfortunately the writing and the figures are quite unclear.

1. *The fabrication process is not clear. For example, Figure S2: how do the authors make sure that the Cr/Au electrodes are touching the graphene and graphite? It is not clear from Figure (h)-(i) at all.*

Our reply: Cr/Au electrodes were placed only in the region where graphite or graphene encapsulated

by hBN exists. Without Cr/Au electrodes contacting graphene or graphite, electrical paths between Cr/Au contacts should be open. According to Science, 342, 6158, 614-617, 2013, when hBN/graphene/hBN is etched and Cr/Au is deposited, Cr/Au forms one-dimensional edge contact to the graphene as shown in the Figure below cropped from the reference. We followed the same procedure to make edge contacts to graphene and graphite by using plasma etching and depositing Cr/Au. To make the edge contact to graphene clarified, we added “One-dimensional edge contact on graphene was formed in this process¹.” In the Supplementary Section 2, Device Fabrication.

2. Figure 1b: What is VTG held at during this measurement? At $V_{BG} = -30$ V, is the MoS₂ still ON? At $V_{BG} = -30$ V, holes are dominant in graphene. So holes should move from graphene to graphite through MoS₂. The authors mention that electrons are being injected from graphene to graphite. Please elaborate.

3. Figure 2: Again, what is VTG here?

Our reply: We appreciate the Referee mentioning the point. To decrease noise level (off state current level in diode measurement) from ~ 50 pA to < 10 fA (three orders of magnitude lower noise level), we newly set up the measurement equipment by replacing all the wires with triaxial cable. We perform additional experiment with additional MoS₂ device. We replace the figure with better data and add **all the information including all the top gate voltages in the figure and figure caption**. We also add two probe gate-dependent IV curve of MoS₂ channel at fixed bias voltage applied to graphene in Supplementary Figure S3 (newly added figure) to elucidate when the MoS₂ channel becomes on

and off (threshold voltage). In the older version of Figure 3 in previous manuscript, graphene was p-type at $V_{BG} = -30V$. Main Fig. 1 shows that as negative bias voltage is applied to graphene, the diode becomes forward bias state (on state). At this *negative bias* regime, electrons move (injects) from graphene to graphite, while holes injects from graphite to graphene. Please note that hole injection is the same of electron injection.

4. Figure 3: I am not sure how the authors are claiming that an ideality factor of less than 1 is sustained for >2 decades. By the green line drawn, it is clear that ideality factor of less than 1 is maintained for a little over 1 decade for $V_{BG} = -15 V, -30 V$ and $-45 V$.

Our reply: The **red** (not green in the previous manuscript) line indicates ideal diode curve with ideality factor = exactly 1. The **green** line indicated average ideality factor over one decade of drain current that we observed. To avoid confusion, we denote the average ideality factor over 1 decade as η_{ave_1dec} . In the Figure 3 of new manuscript (from additional experiment with additional device) clearly shows that average ideality factor is less than 1 over more than 3 decades of drain current since the average slope over 3 decade of current is steeper than the ideal diode curve with ideality factor = 1.

5. How repeatable are the I-V measurements? Will the ideality factor be retained for 10 consecutive sweeps? Please include comments in the manuscript.

Our reply: We thank the Referee for bringing up this point. In our additional device, we performed the suggested measurement. We have found that 10 consecutive measurements shows nearly same IV curves with the nearly same ideality factor $\eta_{\text{ave_1dec}}=0.84$. We added the repeatability of the device into Supplementary Figure S16, and Supplementary section 14.

6. Will the device behave the same way if one were to use CVD-grown MoS₂ and graphene? Please include comments in the manuscript.

Our reply: We thank the Referee for bringing up this point. We have changed the last sentence in the conclusion paragraph from “The realisation of a steep-slope DS diode paves the way for the development of low-power circuit elements and energy-efficient circuit technology.” to “By using CVD-grown MoS₂, graphene and graphite, integrated circuits using steep slope DS-FETs and DS-diodes can be fabricated in a large scale and pave the way for energy-efficient circuit technology.”

Reviewer #3 (Remarks to the Author):

The manuscript entitled “Dirac-Source Diode with Sub-unity Ideality Factor” have demonstrated a Dirac Source diode, which exhibits a steep-slope characteristic curve by using graphene electrodes. This topic and their findings are very interesting. However, one of the major problems is the large leakage current for I_{ds} - V_{ds} measurement. As shown in Fig. 1b and Fig. 3, the I_{ds} current cross zero (y-axis) at very negative V_{ds} voltage, indicating the leakage current have been coupled into the I_{ds} measurement. Therefore, the measured I_{ds} may not truly reflect the diode behavior . This could be important because the I_{ds} current (of sub-1-ideality factor region) is only one order higher than the leakage current. To ensure the accurate ideality factor, lower leakage current (below pA) or higher resolution equipment need to be used, which should be satisfied by standard semiconductor analyzer.

Similarly, if we look into the data in Fig. S8a, b, the top gate leakage current also have a strong influence to the off-state current, which could be another reason for the measured $SS < 60$ mV/dec (which also happens at the device off state). Hence, this top gate leakage should also be minimized to support the author’s claim.

Our Reply: We thank the Referee for pointing out a way to improve the manuscript. To decrease noise level (off state current level in diode and FET measurement) from ~ 50 pA to < 10 fA (more than three orders of magnitude lower noise level), we have newly set up the measurement equipment by replacing all the measurement cables with triaxial cables. We perform additional experiment with additional MoS₂ devices. We have achieved ideality factor less than 1 over 4 decades of drain current as well as $SS < 60$ mV/dec over 4 decades of drain current. We have replaced previous manuscript with a new version having new data with low leakage current level down to 10 fA in the Main and Supplementary Figures.

Based on the present data, I can not support for its publication for Nature Communications. There are some other minor questions as below

1. The device optical image should be included in Fig. 1.

Our reply: We appreciate the Referee mentioning the point. We have added the optical image of our DS diode device in the main Fig. 1.

2. The diode has a local top gate, but the fabrication processes are not described in the manuscript.

Our reply: We have a phrase “Additional e-beam lithography and deposition processes are performed to facilitate top-gate placement.” in the Supplementary section 2. However, the fabrication process was missing in the Figure S2 showing device fabrication procedure. Now, we have added the additional final step of depositing top gate in the supplementary Figure S2 j) as below.

3. At negative back gate voltage, large Schottky barrier should exist between p-type graphene and n-type MoS₂, as claimed by the author “When a negative back-gate voltage is applied, the Schottky barrier height increases”. However, the author also claim the device is dominated by the barrier at graphite and MoS₂: “the device current is mainly modulated by the Schottky barrier at the interface between the graphite and monolayer MoS₂”. This point should be more clarified.

4. Similar as the previous question, the diode behavior only exists at negative back gate voltage according to Fig. 2. Because the graphene/MoS₂ barrier is more sensitive to the gate voltage while graphite/MoS₂ barrier is nearly insensitive to the gate voltage, does this behavior suggest the observed rectifier behavior is more governed by the graphene/MoS₂ rather than the proposed graphite/MoS₂ junction?

Our reply: We agree that at large negative gate voltages, Schottky barrier also forms between p-type graphene and n-type MoS₂. However, this Schottky barrier at graphene/MoS₂ interface is much smaller than the Schottky barrier at graphite/MoS₂, which is implied by IV curve measurement in MoS₂ devices contacted by graphene contacts or by graphite contacts (please see the Supplementary Figure S6).

The domination of Schottky barrier at the interface between graphite and MoS₂ rather than graphene /MoS₂ in the asymmetrically contacted graphene-MoS₂-graphite diode could also be seen in the bias dependent IV curve at large negative bias voltage applied to graphene contact with graphite contact at ground. We apply bias voltage always to graphene as shown in Figure 1 while graphite is grounded. If graphite/MoS₂ Schottky barrier dominates, on current should appear at positive bias voltage rather than

negative bias voltage at negative back gate voltages since for n-type Schottky barrier, positive forward bias should be applied to electrode (see the figure below). Since the diode becomes on (forward bias condition satisfied) when we apply negative bias to graphene with graphite grounded, the Schottky barrier is dominated at the interface of graphite/MoS₂, not graphene/MoS₂. Now, we have added a Supplementary Section 8 and Supplementary Figure S10 to explain why the measured diode IV curves show the dominance of Schottky barrier at the interface of graphite/MoS₂ rather than graphene/MoS₂.

5. For band diagram in Fig. 2b, the graphene should have large Schottky barrier with MoS₂. For diagram in Fig. 2c, if the MoS₂ is degenerated, the band bending is confusing at graphite part. What is the work function of graphite used here?

Our reply: Graphite work function used here is 4.7eV (Rut'kov, E. V., Afanas'eva, E. Y. & Gall, N. R. Graphene and graphite work function depending on layer number on Re. *Diamond Relat. Mater.* **101**, 107576 (2020), Yan, R. *et al.* Determination of graphene work function and graphene-insulator-semiconductor band alignment by internal photoemission spectroscopy. *Appl. Phys. Lett.* **101**, 022105 (2012)). Although we agree that Schottky barrier is formed at graphene/MoS₂, the Schottky barrier at the interface of graphite/MoS₂ dominates the device as we explained in the answer to the question 3 and 4. Therefore, to avoid confusion, we did not include the Schottky barrier at graphene/MoS₂ in the figure.

REVIEWER COMMENTS

Reviewer #1 (Remarks to the Author):

I appreciate authors' efforts to answer my questions. Most of the questions have been addressed adequately by either performing new experiments or simulations. For those that the authors cannot address, explicit reasons are provided.

Overall the quality of the manuscript is improved, and the presentations are more clear than the original one. I can recommend publication of this paper in Nature Comm.

Reviewer #2 (Remarks to the Author):

My concerns have been addressed sufficiently by the authors. I recommend the manuscript for publication.

Reviewer #3 (Remarks to the Author):

In the revised manuscript, the author has made efforts to improve the manuscript quality. However, the major questions about the $SS < 60$ meV/dec and data interpretation still remains, as explained below. Therefore, I can not support for its publication.

As shown in Fig. S12, the $SS < 60$ meV/dec only exist below 3×10^{-14} A, and is very close to gate leakage current. Therefore, the gate leakage current (either positive or negative value) may be coupled into the measurement I_{ds} , leading to underestimated SS . Although the 4-decades SS is also below 60 meV/dec, the actually $SS < 60$ region is only less than 0.5-decade, which could mislead readers. Similarly, for the ideal factor < 1 (presented in Fig. 3), the I_{ds} current is still influenced by the gate leakage current (because the < 1 region have very small current and can be easily influenced). Therefore, the represented data here can not support the main conclusion of this manuscript.

Another minor point is the band diagram. Lots of literatures have theoretically and experimentally suggested graphene is gate tunable and p-type graphene (especially under negative gate voltage) would form large Schottky barrier with MoS₂. For example, if negative gate voltage applied, the Fermi level of graphene is lower than 4.7 eV of graphite. The author should explain why small graphene Schottky barrier is shown in the band diagram in Fig. 2.

Reviewer #3 (Remarks to the Author):

In the revised manuscript, the author has made efforts to improve the manuscript quality. However, the major questions about the $SS < 60$ meV/dec and data interpretation still remains, as explained below. Therefore, I can not support for its publication.

1. As shown in Fig. S12, the $SS < 60$ meV/dec only exist below 3×10^{-14} A, and is very close to gate leakage current. Therefore, the gate leakage current (either positive or negative value) may be coupled into the measurement I_{ds} , leading to underestimated SS . Although the 4-decades SS is also below 60 meV/dec, the actually $SS < 60$ region is only less than 0.5-decade, which could mislead readers. Similarly, for the ideal factor < 1 (presented in Fig. 3), the I_{ds} current is still influenced by the gate leakage current (because the < 1 region have very small current and can be easily influenced). Therefore, the represented data here can not support the main conclusion of this manuscript.

Our Reply: The reviewer wrote that $SS < 60$ and ideality factor < 1 is only within very small range of current near the leakage current. The reviewer wrote that $SS < 60$ mV/dec is only less than 0.5 decade of drain current. To show that this is not consistent with our data, we plot (below) drain current versus ideality factor for the data shown in main Figure 3 c) and drain current versus SS for the data shown in Supplementary Figure 12. Here, both ideality factor < 1 and $SS < 60$ mV/dec holds at least for 2 decade of drain currents from the leakage current. $SS < 60$ mV/dec holds from 10^{-14} A to at least 10^{-12} A. Also, the ideality factor < 1 holds from 10^{-14} A to over 2×10^{-12} A, which is more than 2 decades of drain current. Since the figure of merit for low-power transistor is known to be average SS over 4 decades of current, we measured the average SS over 4 decades of current as defined in the literatures and confirmed that it is below 60 mV/dec. Also, average ideality factor over 4 decades of current is less than 1. **Please note that ideality factor < 1 does not guarantee that $SS < 60$ mV/dec since SS can be further improved by decreasing the thickness of gate dielectric materials.** Ideality factor < 1 holds for more than two decades of drain currents cannot be explained by the leakage current, since the leakage current will affect the drain current region very close to the leakage current level (maybe within 0.5 decades of current or so) as the reviewer pointed out. Therefore, we believe our data of ideality factor less than unity over more than two decades of current and average ideality factor over four decades of current less than one support our main conclusion.

2. Another minor point is the band diagram. Lots of literatures have theoretically and experimentally suggested graphene is gate tunable and p-type graphene (especially under negative gate voltage) would form large Schottky barrier with MoS₂. For example, if negative gate voltage applied, the Fermi level of graphene is lower than 4.7 eV of graphite. The author should explain why small graphene Schottky barrier is shown in the band diagram in Fig. 2.

Our Reply: As shown in Figure S6, graphene-MoS₂ shows nearly Ohmic contact differently from graphite-MoS₂ contact. We wrote in the manuscript, “Without gate modulation, graphene has a work function of 4.3–4.7 eV from a monolayer to a few layers²³⁻²⁵. Because the work function of graphene (~4.3 eV) does not differ significantly from the electron affinity of MoS₂ (~4.2 eV)²⁶⁻²⁹, the Schottky barrier height at the graphene/MoS₂ interface is negligible, compared to the Schottky barrier height at the graphite/MoS₂ interface. This also indicates that the Dirac point of pristine graphene is located near the conduction band edge of MoS₂. As shown in Fig. S10, in case of metal/n-type semiconductor junction, positive voltage on metal became forward bias. In our case, we applied bias voltage on graphene side, and negative bias became forward bias, i.e., positive bias on graphite side is forward bias, which indicates Schottky barrier between the graphite/MoS₂ junction is dominated in our device. Fig. S6 indicates that the graphene/MoS₂ device shows an almost Ohmic IV curve, whereas graphite/MoS₂ does not show an Ohmic IV curve at room temperature. “ in line from 114 to 123. Our result is consistent with literatures reported so far. For example, please see Liu, Y. *et al.* Toward Barrier Free Contact to Molybdenum Disulfide Using Graphene Electrodes. *Nano Lett.* **15**, 5, 3030-3034 (2015). [10.1021/nl504957p](https://doi.org/10.1021/nl504957p). The data from the paper is plotted below. At room temperature, monolayer MoS₂ contacted by graphene shows Ohmic behavior, which is consistent with our result.

Figure 1

Figure 1. Device schematics and structural characterizations. (a) Perspective and cross-sectional schematics of MoS₂ device structure with bottom-graphene electrodes. (b) Optical image of two graphene strips placed close to each other (top panel) and the final device after MoS₂ transfer and metal electrode deposition (bottom panel). (c) Cross-sectional TEM image of the graphene–MoS₂ interface, indicating the ultraclean and sharp interface resulted from the dry transfer technique.

Figure 2

Figure 2. Output characteristics of MoS₂ devices contacted by graphene. (a,b) Output characteristics of a monolayer MoS₂ device at room temperature (a) and low temperature (1.9 K) (b) at varying gate voltages. Linear I - V behavior is observed in both cases, particularly at high positive gate voltages. Gate voltages range from -60 to 80 V with a 20 V step. (c,d), Output characteristics of a multilayer MoS₂ device at room temperature (c) and low temperature (1.9 K) (d). Linear I - V behavior is observed in both cases, particularly at high positive gate voltages. Gate voltages range from -60 to 60 V with a 20 V step.

REVIEWER COMMENTS

Reviewer #1 (Remarks to the Author):

The reviewer #3 did raise some legitimate concerns about the extent to which the leakage current affects the ideality factor and sub-60mV/dec SS. To fully address reviewer #3's question, the authors should probably present the leakage current under the measurement conditions. However, I understand that for the presented devices it may be difficult as the gate leakage may not have been recorded during the measurements. In that case results of new devices may be presented as supplementary material to demonstrate the robustness of the sub-1 ideality factor. For the questions regarding the band alignment, I believe the authors made a reasonable explanation.

Reviewer #2 (Remarks to the Author):

I think that the authors' justifications are believable and I recommend the manuscript for publication.

The SS below 60 mV/dec is sustained for 4 decades of current, even though the current values are low. Upon changing cables, the leakage current that is measurable has decreased, which is expected. Hence, the SS < 60 mV/dec range has decreased.

About the band diagram, I agree with the authors that graphene/MoS₂ barrier can be low enough to be ohmic (<https://doi.org/10.1021/nn501723y>). Hence I agree with the authors' representation of the band diagrams.

Reviewer #3 (Remarks to the Author):

Although the author did a quick response to my previous concerns, their explanation is not solid and I can not support its publication without addressing these key questions, as explained below.

In the original manuscript of Fig. S8 (I attached below), the $SS < 60$ mV/dec region first emerges at I_{ds} current level ~ 100 pA to 1 nA (10^{-10} A to 10^{-9} A), which is very close to the leakage current of the measurement system. I raised this question in my previous comments, and thanks to the authors, they have realized this problem and replaced all the measurement cables with triaxial cables with higher resolution (as shown in the revised Fig. S12, and attached below). However, after improving the measurement setup, the $SS < 60$ mV/dec did not show up again at $I_{ds} \sim 100$ pA to 1 nA, and author did not explain why this happens. Instead, it first emerges at much lower current level ~ 0.03 pA (3×10^{-14} A), as highlighted in the attached figure below), which is also close to the leakage current of the system.

Since the $SS < 60$ mV/dec region is always at the same range with the leakage current no matter what measurement setup used, the explanation of the working mechanism is not solid, and it implies that the leakage current impacts the SS extraction. Without solving this key question, the data presented can not support the authors claim and I can not support the manuscript for publication.

For the question regarding to the band diagram. In the author mentioned reference (Nano Lett. 15, 5, 3030 (2015)), they indicate the graphene is highly gate tunable and Ohmic contact only forms at large positive gate voltage (+80 V for monolayer and +60 V for 20 layer device, as shown in authors response letter). At negative gate voltage, large Schottky actually forms between graphene and MoS₂. The Schottky barrier (at negative gate voltage) is large enough to dominate the whole carrier transport, and the large barrier actually enables a new field of vertical transistors to switch the device off [Appl. Phys. Lett. 105, 083119 (2014); Appl. Phys. Lett. 106, 223103 (2015); Nat. Mater. 12, 246–252 (2013)]. Therefore, the author should carefully label the energy of each part (graphene at positive/negative gate, graphite, MoS₂ conduction band and valence band) in their band diagram in Fig. 2 to analysis the band alignment and clarify this question. Especially the band is not bending at all at graphene side, regardless graphene Fermi level difference between Fig.2b and 2c.

Reviewer #1 (Remarks to the Author):

The reviewer #3 did raise some legitimate concerns about the extent to which the leakage current affects the ideality factor and sub-60mV/dec SS. To fully address reviewer #3's question, the authors should probably present the leakage current under the measurement conditions. However, I understand that for the presented devices it may be difficult as the gate leakage may not have been recorded during the measurements. In that case results of new devices may be presented as supplementary material to demonstrate the robustness of the sub-1 ideality factor. For the questions regarding the band alignment, I believe the authors made a reasonable explanation.

Our reply to Reviewer 1:

We greatly thank the reviewer for giving us advices on how to address the concerns raised by referee #3 and to improve our manuscript further. As the referee points out, we have measured the leakage current of the system and the gate, which was 2-5fA (please see the red dots below). On the other hand, reverse bias current in diode measurement and off-current in field-effect transistor measurement were measured to be ~50fA and ~20fA, respectively, which are much larger than the leakage current of the measured system and gate leakage. Now we have added the leakage current measured along with currents through device in Supplementary Figure S17.

Our first version of manuscript showed high leakage current (~50pA), and we fully understood there was concern about high leakage current. Because the reverse bias current in

diode measurement and off-current in field-effect transistor measurement were the same with leakage current of measurement system ($\sim 50\text{pA}$), the measured off-current and reverse bias current could not represent the actual off-current and reverse bias current of the measured device (please see Figure below). The figure below shows leakage current (red) and the current we measure in the device (black). As the leakage current of the system is very high $\sim 50\text{pA}$, the off-state current is dominated by this system leakage current.

In the case when system leakage dominates the off-state current, IV curve can be interpreted in a wrong way in the range of current near the off-state current level. For example, when we measured (displayed in the measurement instruments) forward current of -50pA and -500pA ($\times 10=1$ decade), the actual current flow through the device can be regarded as -100pA and -550pA ($\times 5.5=0.74$ decade) due to the $+50\text{pA}$ of leakage current. We fully understood that the original data had a problem due to this large system leakage current and therefore we performed measurement with new system having much lower system leakage current in a new device.

Considering the same interpretation as in the case of high leakage current, when the measured reverse bias- and off-current level is much higher than the leakage current, forward bias- and on-current near the off-state current level can represent **the actual current** flowing in the device. For example, when we measured (displayed in the measurement instruments) forward current of -50fA and -500fA ($\times 10=1$ decade), the actual current flow through the device can be regarded as -52fA and -502fA ($\times 9.65=0.98$ decade) due to the $+2\text{fA}$ of leakage current, which is almost similar with measured data. Therefore, we claim that our data measured with ultra-low leakage current ($2\text{-}5\text{fA}$) support the steep-slope behavior in diode and field-effect transistor

above the leakage current level.

Our reply to Reviewer 3:

- We thank the reviewer for pointing out the important concern and the way to improving our manuscript. The reviewer pointed out concerns about the change in the current level at which the steep slope behavior occurs in field-effect transistor measurement between the original manuscript and the re-submitted manuscript. In the first round of review, the reviewer raised concern about high leakage current ($\sim 50\text{pA}$), and we fully understood the issue. Because the reverse bias current in diode measurement and off-current in field-effect transistor measurement were the same with leakage current of measurement system ($\sim 50\text{pA}$). As reviewer pointed out, measured off-current and reverse bias current cannot represent the actual off-current and reverse bias current of the measured device (please see below figure 1). The figure below shows leakage current (red) and the current we measure in the device (black). As the leakage current of the system is very high $\sim 50\text{pA}$, the off-state current is dominated by this system leakage current.

In the case when system leakage dominates the off-state current, IV curve can be interpreted in a wrong way in the range of current near the leakage current level. For example, when we measured (displayed in the measurement instruments) forward current of -50pA and -500pA ($\times 10=1$ decade), the actual current flow through the device can be regarded as -100pA and -550pA ($\times 5.5=0.74$ decade) due to the $+50\text{pA}$ of leakage current. We fully understood that the original data had a problem due to this large system leakage current and therefore we performed measurement with new system having much lower system leakage current in a new device.

In the newly measured data with tri-axial cables, the leakage current of measurement system and gate was measured about 2~5fA (see red dots in Figure 2 below), which is reduced more than 3-orders of magnitude compared with previous measurement system. On the other hand, reverse bias current in diode measurement and off-current in field-effect transistor measurement were measured to be ~50fA and ~20fA, respectively, which are much larger than the leakage current of the measured system and gate leakage. Now we have added the leakage current measured along with currents through device in Supplementary Figure S17.

Considering the same interpretation as in the case of high leakage current, when the measured reverse bias- and off-current level is much higher than the leakage current, forward bias- and on-current near the off-state current level can represent **the actual current** flowing in the device. For example, when we measured (displayed in the measurement instruments) forward current of -50fA and -500fA ($\times 10=1$ decade), the actual current flow through the device can be regarded as -52fA and -502fA ($\times 9.65=0.98$ decade) due to the +2fA of leakage current, which is almost similar with measured data. Therefore, we claim that our data support the steep-slope behavior in diode and field-effect transistor above the leakage current level.

To avoid confusion and noise in steep-slope field-effect transistor measurement (revised Fig.S12), we re-plot the original graph with 10mV steps which was previously 2mV steps with noise (please see below, revised Fig.S12).

- Regarding the band diagram, we have observed large Schottky barrier which is also consistent with literature (*ACS Nano* 2014, 8, 6, 6259, *Nano Lett.* **15**, 5, 3030-3034 (2015). [10.1021/nl504957p](https://doi.org/10.1021/nl504957p)). The Nano Letter reports Ohmic contact behavior at both positive and negative gate voltages as shown below ($V_g = -60V$ to $80V$). From our measurement (Supplementary Figure S6), we observed Ohmic contact (meaning no significant barrier causing nonlinear IV curves at room temperature) formed between monolayer MoS₂ and graphene at all gate voltages. We understand that other literatures report large Schottky barriers, so this contact problem needs further studies.

Figure 2

Figure 2. Output characteristics of MoS₂ devices contacted by graphene. (a,b) Output characteristics of a monolayer MoS₂ device at room temperature (a) and low temperature (1.9 K) (b) at varying gate voltages. Linear $I-V$ behavior is observed in both cases, particularly at high positive gate voltages. Gate voltages range from -60 to 80 V with a 20 V step. (c,d) Output characteristics of a multilayer MoS₂ device at room temperature (c) and low temperature (1.9 K) (d). Linear $I-V$ behavior is observed in both cases, particularly at high positive gate voltages. Gate voltages range from -60 to 60 V with a 20 V step.

REVIEWERS' COMMENTS

Reviewer #1 (Remarks to the Author):

The authors provided a satisfactory response to my previous questions. I'd like to recommend publication of this paper in Nature Comm.

Reviewer #3 (Remarks to the Author):

The author have addressed my previous question of gate leakage current. But for the other question about the band alignment, they did not provide enough explanation about the Ohmic contact (or negligible barrier between graphene) under negative enough gate voltage. I would to point it out in the mentioned reference (Nano Lett. 15, 5, 3030 (2015)) or your measurement (Fig. S6), the Ohmic contact is not observed under negative gate voltage, if you plot the I_{ds} - V_{ds} at -60 V gate along.

Since most questions are well-answered, I could support it to publish, and leave the bandgap question for further investigation.